# Vision transformer with masked autoencoders for referable diabetic retinopathy classification based on large-size retina image

Yaoming Yang[1], Zhili Cai[1], Shuxia Qiu[1,2], Peng Xu[1,2]*

**1** College of Science, China Jiliang University, Hangzhou, Zhejiang, China, **2** Key Laboratory of Intelligent Manufacturing Quality Big Data Tracing and Analysis of Zhejiang Province, Hangzhou, Zhejiang, China

* xupeng@cjlu.edu.cn

**Data Availability Statement:** All data are available from: EyePACS:https://www.kaggle.com/competitions/diabetic-retinopathy-detection/data APTOS:https://www.kaggle.com/competitions/

## Abstract

Computer-aided diagnosis systems based on deep learning algorithms have shown potential applications in rapid diagnosis of diabetic retinopathy (DR). Due to the superior performance of Transformer over convolutional neural networks (CNN) on natural images, we attempted to develop a new model to classify referable DR based on a limited number of large-size retinal images by using Transformer. Vision Transformer (ViT) with Masked Autoencoders (MAE) was applied in this study to improve the classification performance of referable DR. We collected over 100,000 publicly fundus retinal images larger than 224×224, and then pre-trained ViT on these retinal images using MAE. The pre-trained ViT was applied to classify referable DR, the performance was also compared with that of ViT pre-trained using ImageNet. The improvement in model classification performance by pre-training with over 100,000 retinal images using MAE is superior to that pre-trained with ImageNet. The accuracy, area under curve (AUC), highest sensitivity and highest specificity of the present model are 93.42%, 0.9853, 0.973 and 0.9539, respectively. This study shows that MAE can provide more flexibility to the input image and substantially reduce the number of images required. Meanwhile, the pretraining dataset scale in this study is much smaller than ImageNet, and the pre-trained weights from ImageNet are not required also.

## Introduction

According to the WHO report, there are more than 400 million people with diabetes in the world [1]. The number of people living with diabetes is projected to reach 552 million by 2024 [2]. DR caused by diabetes is one of the leading causes of blindness, which is probably avoidable [3]. However, the screening of DR involves many features and it is time-consuming for clinicians [4]. In addition, the diagnosis results will be largely affected by the doctor's personal work experience, professional standards, psychological state, and other factors. Thus, computer-aided techniques have been proposed to improve the accuracy and enhance the efficiency of DR diagnosis [5].

aptos2019-blindness-detection/data Messidor2:
https://www.adcis.net/en/third-party/messidor2/
OIA:https://github.com/nkicsl/OIA The Code is
available at: https://github.com/CNMaxYang/
VMLRI.

**Funding:** This work was supported by the National
Natural Science Foundation of China through grant
number 52376079. The URL for NSFC is https://
www.nsfc.gov.cn/. The funders had no role in
study design, data collection and analysis, decision
to publish, or preparation of the manuscript.

**Competing interests:** The authors have declared
that no competing interests exist.

Since the theory of deep learning was proposed in 2006 [6], it has been developed with the enhancement of computational power [7]. Due to the excellent ability of feature extraction and classification, deep neural networks have been introduced in the medical field [8–11] and have been successfully applied in DR diagnosis [12–14]. Compared with the state-of-the-art CNN architecture, ViT shows better performance on classification tasks of computer vision (CV) [15–17] and therefore has been applied in various downstream CV tasks [18].

Recently, Kumar et al. [19] tested the classification performance of several major CNNs and Transformers as well as MLPs on APTOS dataset. They found that Transformers perform better than CNNs and MLPs overall. However, Dosovitskiy et al. [17] pointed out that the sufficient training on a large number of images should be carried out in ViT due to the lack of inductive bias. Touvron et al. [20] introduced a new token-based distillation strategy based on ViT and named the model as DeiT (Data efficient image Transformers) to reduce the data requirements of ViT. Matsoukas et al. [21] tested the classification performance of DeiT-S on APTOS and found that it is only close to that by ResNet50 when the pre-training data is limited.

He et al. [22] proposed MAE to overcome the issue that the training of Transformer requires a lot of data, they achieved 87.8% accuracy on ImageNet using MAE and ViT-Huge. The cost of acquisition and annotation of medical images is much higher than that of natural images. And the size of medical images is generally larger than 224×224, which is much higher than that of natural images. Although the self-attention mechanism in ViT can handle sequences of any length, the pre-trained position embeddings cannot be directly applied on images larger than 224×224. And Shamshad et al. [18] argued that the pre-trained weights from ImageNet are not optimal for medical images. Srinivasan et al. [23] further pre-trained their model using the EyePACS dataset in a self-supervised manner on top of the ImageNet weights. They observed that pre-training with retinal images could further enhance the model's classification performance and mitigate overfitting. This finding suggests that, to some extent, ImageNet pre-trained weights may not be the optimal choice for diabetic retinopathy classification.

The rest of this paper is organized as follows. The second part briefly describes the related work. The third part introduces the datasets and methods used in this study. The fourth part illustrates the experiment details and results. The fifth part contains the conclusion and prospect.

## Related work

The Transformer was first proposed in 2017 and then successfully applied on natural language processing (NLP) [24]. Then, Radford et al. [25] proposed GPT based on Transformer in 2018, and Devlin et al. [26] presented BERT and got state-of-the-art results in 11 NLP tasks in 2019. With the success of Transformer in NLP, ViT was proposed and applied in CV by Dosovitskiy et al. in 2020 [17].

### Transformer

Yang et al. [27] used the sliding window method to extract patches in order to avoid the key area of the lesion being divided into different patches. They employed CNN to reduce the dimension of the patches, and inputted the reduced patches to ViT. They also selected the patches with the largest weight as the effective area by accumulating the attention weight. They utilized the OIA-ODIR dataset with image resolution of 224×224 as input and achieved an accuracy of 84.1%. Jha et al. [28] attempted to classify multiple diseases including DR on OCT B-scan data. They employed images with a resolution of 256×256 as input, the accuracy of ViT

and VGG-16 are 88% and 83%, respectively. They also proposed a hybrid structure of ViT and SVM. The hybrid structure also utilizes images with a resolution of 256×256 as input, achieving an accuracy increase to 94%. In addition, the combination of ViT with CNN was also proposed and applied [29, 30]. Sadeghzadeh et al. [29] fused EfficientNet-B0 with Transformer and achieved state-of-the-art classification results using 224×224 images as input on EyePACS, APTOS, DDR, Messidor-1, and Messidor-2 datasets. Ma et al. [30] proposed a fusion network based on Transformer and CNN for the grading of DR, where DR grading was treated as a joint ordinal regression and multi-classification problem. They utilized images with a slightly higher resolution of 384×384 and demonstrated superior performance on the DeepDR and IDRiD datasets. Adak et al. [31] integrated four Transformer models, ViT, BEiT (Bidirectional Encoder representation for image Transformer), CaiT (Class-Attention in Image Transformers) and DeiT, for DR detection. Based on the APTOS dataset with image resolution of 256×256, they indicated that the integration of multiple Transformer models further enhances the model's classification capabilities for DR.

## MAE

Recently, Masked Image Modeling (MIM) in the self-supervised learning field has been further developed [32–34]. Based on ImageNet dataset, He et al. [22] applied MAE in ViT-Huge to achieve the accuracy of 87.8%. Encouraged by this result, researchers began to explore the application of MAE in medical images. Zhou et al. [35] took ViT-Base as a backbone and used MAE in the pretraining phase, where ChestX-ray14, BTCV and BRATS datasets were tested. The image resolutions of the ChestX-ray14, BTCV, and BraTS dataset are 224×224, 96×96×96, and 128×128×128, respectively. They reported that the AUC was increased by 9.4% for classification tasks, and the average DSC can be improved from 77.4% to 78.9% for the tumor segmentation task. Cai et al. [36] generated a multi-modal and multi-dimensional dataset with 95,978 samples, and named it as mmOptht-v1. They also designed a general architecture (UnionEye), which shows good performance on both 2D (resolution of 224×224) and 3D (resolution of 112×224×112) image-related tasks. In addition, they found that the model with a masking ratio of 50% was better than that with a masking ratio of 75% in terms of AUC, Recall, Kappa, and other metrics.

## Our contributions

In order to improve the performance of Transformers on large-size medical images, over 100,000 available large-size fundus retinal images were used to pre-train ViT by MAE in the present work, it was then fine-tuned and tested on APTOS dataset with labels. The structure of proposed VMLRI is shown in Fig 1. It is divided into two parts: pre-training part and fine-tuning part. In the pre-training phase, inputted retinal images are uniformly splitted into non-overlapping image blocks (patches). Subsequently, a portion of these image blocks is masked to create mask tokens based on a predetermined masking rate (e.g., 75% or 50%). The remaining visible image blocks are then encoded by the ViT. Upon completion of encoding, mask tokens are introduced into the encoded results, and together they are fed into a simple decoder to attempt the reconstruction of the original image. After pre-training, the decoder component is discarded, the ViT can be utilized for the classification of referable DR. To the best of our knowledge, this is the first study that employs MAE to pre-train large-size fundus retinal images for referable DR (rDR) classification.

The objective of this study is to reduce the number of images required for pre-training ViT in the DR domain through MAE and to achieve superior classification performance for DR compared to ViT pre-trained with over one million natural images. Thus, the proposed model

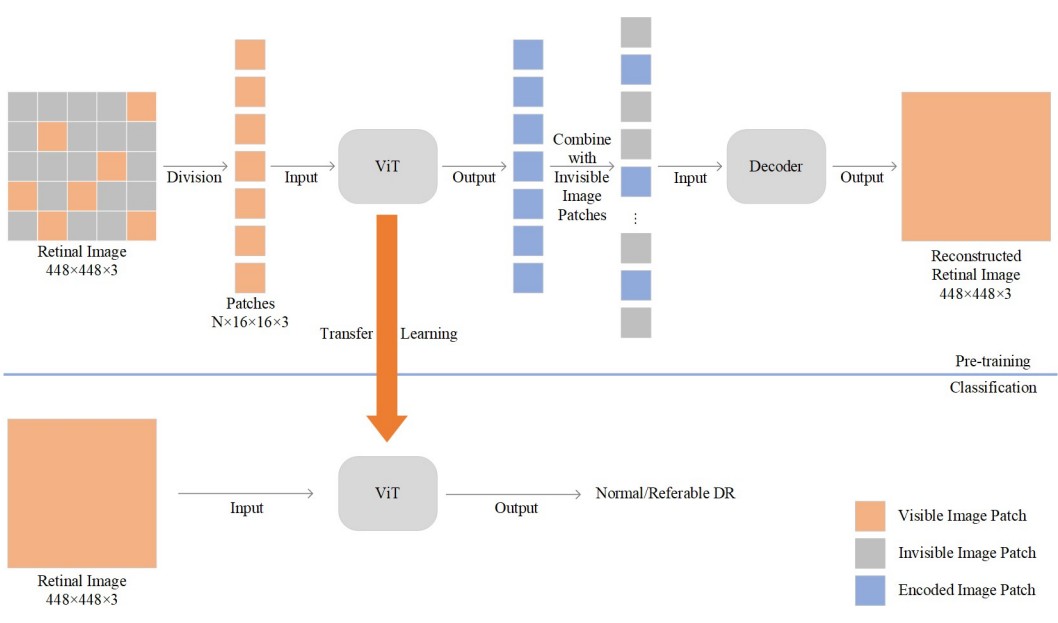

**Fig 1. The architecture of ViT with MAE based on large-size retina image (VMLRI).**

pre-trained on different sizes of retinal images is also compared with the ViT that pre-trained on ImageNet. Since the inter-class gap of fundus retinal images is smaller than that of natural images, a masking rate of 50% is also tested and compared with that of 75% masking ratio.

## Materials and methods

### Datasets

The APTOS [37] public dataset contains 5,590 color fundus retinal images, but only 3,662 images have corresponding DR grading labels. The kaggle-EyePACS public dataset [38] contains 88,702 color fundus retinal images, of which 35,126 images have corresponding DR grading labels. DR was graded in this dataset in the same way as that of APTOS. Messidor-2 [39] is an extension of the original Messidor dataset. It contains 1,748 color fundus retinal images, but the fundus images do not appear in pairs and the official DR grade for each image is not provided. OIA-DDR [40] is a subset of the OIA series datasets, which contains 13,673 fundus images with labels of DR classification and segmentation. All the data in kaggle-EyePACS and Messidor-2 as well as OIA-DDR, and the data without labels in APTOS were used as pretraining data. And the data with corresponding labels in APTOS was taken as fine-tuning data and testing data. The total number of images used for pretraining is 106,051, and that for fine-tuning and testing is 3,662.

Meanwhile, in order to study the performance gap of MAE on datasets with different scales, three datasets were constructed. Dataset1 contains 17,349 fundus retinal images, it is consisted of unlabeled data in APTOS, Messidor-2 and OIA-DDR. Dataset2 contains 106,051 fundus retinal images, it is composed by Dataset1 and EyePACS dataset. Dataset3 contains 3,662 fundus retinal images, it is formed with labeled data in APTOS. The data information for these three datasets is shown in Table 1.

### Vision transformer and masked autoencoders

A central part of the Transformer architecture is the self-attention mechanism, which was first proposed in CV in 2014 [41] and then was applied in NLP [42]. The self-attention also known

**Table 1. The dataset used in this study.**

| Dataset Name | Compose | Number of images |
|---|---|---|
| DataSet1 | APTOS without labels, Messidor-2, OIA-DDR | 17,349 |
| DataSet2 | APTOS without labels, Messidor-2, OIA-DDR, EyePACS | 106,051 |
| DataSet3 | APTOS with labels | 3,662 |

as scaled dot-product attention, can be expressed by:

$$Attention(Q, K, V) = softmax\left(\frac{QK^T}{\sqrt{d_K}}\right)V \tag{1}$$

where $Q$, $K$ and $V$ represents query, key and value, respectively. These parameters are all obtained by different mapping of the input. Initially, Transformer consists of encoder and decoder, which is composed by several Transformer blocks. While, the Transformer block is mainly composed of residual connection, multi-head attention mechanism, fully connected layer and layer normalization. The input of ViT were replaced by 16×16×3 patches which were obtained via partitioning the original 224×224×3 images.

The structure of ViT is shown in Fig 2, where ViT-Large and ViT-Base are employed. The input image is initially uniformly divided into non-overlapping patches, which are then mapped to vectors of a predetermined dimension through a straightforward linear transformation. These vectors are subsequently added to learnable position vectors of the same dimension. The resulting vectors are fed into Transformer blocks for encoding, and the final classification results for the image are obtained through a simple MLP. The details of ViT-Large and ViT-Base models are listed in Table 2. However, it is lack of the position embedding in the pre-trained weights when the patches increase with the enhancement of

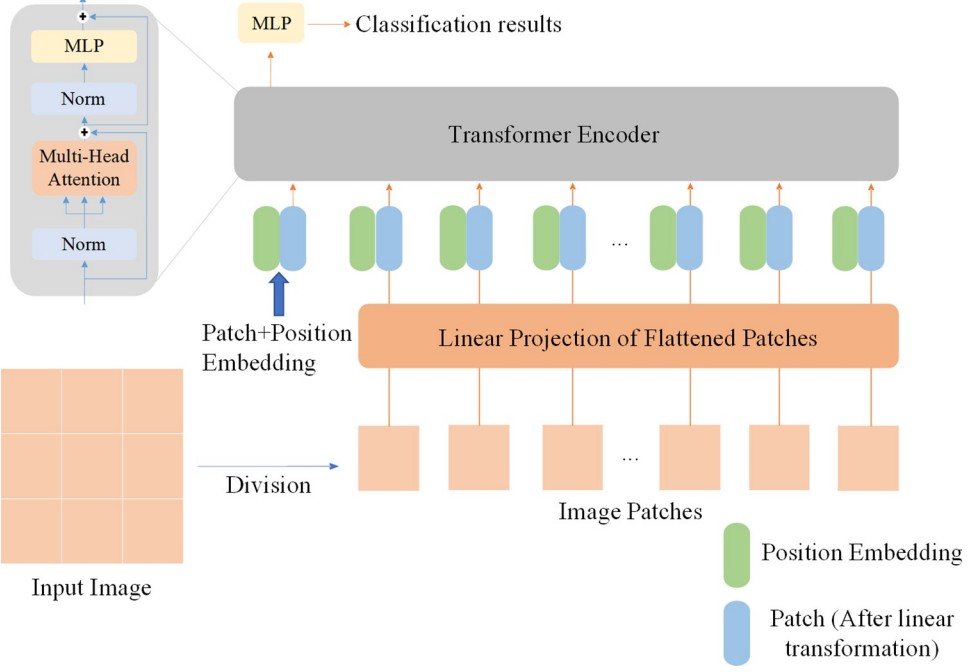

**Fig 2. The architecture of ViT.**

**Table 2. The details of ViT-Large and ViT-Base.**

| Model | Layer | Hidden size D | MLP size | |
|---|---|---|---|---|
| ViT-Base | 12 | 768 | 3072 | |
| ViT-Large | 24 | 1024 | 4096 | |

input image size. Thus, the Bicubic interpolation was used here to obtain the missing position embedding in the pre-trained weights.

MAE is one of simple and scalable self-supervised learning methods. It was used to pretrain ViT-Large and ViT-Base models. The input images were divided into nonoverlapping patches with the same size in the pre-train stage. Then, a high proportion of image patches were randomly masked, and the remaining unmasked image patches were encoded by the encoder. A simple decoder was used to restore the original images based on the output of the encoder and the mask token. When the pretraining process was complete, the decoder part was discarded and the rest of the network can be used for a specific task via transfer learning.

## Experiment and results

### The detail of experiments

In order to make a fair and objective comparison of the experimental results as much as possible, the hyperparameters were fixed for all experiments. The main hyperparameters are summarized in Table 3. Dataset3 was used to finetune and test the pre-trained model. The test goal is the classification performance of the model for rDR. Four evaluation metrics, accuracy, AUC, sensitivity and specificity were recorded in each experiment to evaluate the final performance of different pretrained models. At first, 10% of the images in Dataset3 were randomly selected as the test set. And then the rest of the data was randomly splitted into 80% and 20% parts, which were respectively taken as the training set and the validation set. All the results of this study were obtained by a single test with the model on the test set. After the cropping operation, all images were resized to the same size as the pretraining images. random horizontal flip and random rotation in the range of (-180 degrees, +180 degrees) were performed in the image enhancement. The optimizer was stochastic gradient descent with momentum. The initial learning rate was $1.5625\times10^{-3}$ and the momentum was 0.9. The learning rate was multiplied by 0.8 for epochs 10, 25 and 50. The WeightedRandomSampler in Pytorch was used for class balancing. The binary cross-entropy loss function was used and the maximum number of epochs was 150.

In order to compare with the results of pretraining using fundus images, the randomly initialized weights and ImageNet pre-trained weights were loaded in ViT on Dataset3 with the same hyperparameters. The Bicubic interpolation was also carried out to obtain the lacking position embeddings when the ImageNet pre-trained weights were loaded and then fine-tuned on images larger than 224×224. In the pretraining stage, the Adam optimizer was used for the

**Table 3. Main hyperparameters configured in the experimental setup of this study.**

| | Pre-training (224×224 and 320×320) | Pre-training (448×448) | Finetune |
|---|---|---|---|
| Optimizer | Adam | AdamW | SGD |
| Optimizer momentum | $\beta_1 = 0.9, \beta_2 = 0.999$ | $\beta_1 = 0.9, \beta_2 = 0.95$ | 0.9 |
| Learning rate | $10^{-4}$ | $10^{-4}$ | $1.5625\times10^{-3}$ |
| Learning rate schedule | - | - | milestones |
| Augmentation | random horizontal flip and random rotation | random horizontal flip and random rotation | random horizontal flip and random rotation |

**Table 4. The pretraining information from ViT-Large.**

| Dataset | Image size | Epoch | Masking Ratio |
|---------|-----------|-------|---------------|
| DataSet1 | 224×224 | 100, 200, 300, 400, 500, 600, 700, 800, 900, 1000 | 0.5, 0.75 |
| | 320×320 | 100, 200, 300, 400, 500, 600, 700, 800, 900, 1000 | 0.75 |
| DataSet2 | 224×224 | 100, 200, 300, 400, 500, 600, 700, 800, 900, 1000 | 0.5, 0.75 |
| | 320×320 | 100, 200, 300, 400, 500, 600, 700, 800, 900, 1000 | 0.75 |

224×224 and 320×320 images and the learning rate was fixed to be $10^{-4}$. The AdamW optimizer ($\beta_1 = 0.9$, $\beta_2 = 0.95$) was used to pretrain on 448×448 images, and the learning rate is fixed to be $10^{-4}$. The image processing operations were the same as on Dataset3.

For ViT-Large, a maximum number of pretraining epochs of 1000 was used on Dataset1 and Dataset2, and the pre-trained model was saved every 100 epochs. The masking ratios of 75% and 50% were used on Dataset1 and Dataset2, where the image sizes were all adjusted to 224×224. When the images in Dataset1 and Dataset2 were resized to 320×320, the masking ratios was set to 75%. The pre-training information of ViT-Large is summarized in Table 4.

For ViT-Base, a masking ratio of 75% was used on Dataset1 and Dataset2. The maximum number 800 pretraining epochs was used on 224×224 images, while 1000 maximum pretraining epochs were used on images larger than 224×224 (320×320 and 448×448). The pre-trained model was saved every 100 epochs. The pre-training information in ViT-Base is summarized in Table 5.

## Results and analysis

The results of the ViT-Large experiment with masking ratio of 0.75 are shown in Table 6. It is evident that the model pre-trained on ImageNet shows better results than that with randomly initialized weights. The accuracy of the present model based on 320×320 images of Dataset2 exceeds that pre-trained on ImageNet with Bicubic interpolation. And the ViT-Large pre-trained on 320×320 images of Dataset2 outperforms that pre-trained on ImageNet in AUC and sensitivity. Based on 17,000 images of Dataset1, the accuracy can be increased from 89.59% to 90.41%, AUC can be enhanced from 0.9624 to 0.9719, and the sensitivity can be enhanced from 0.9324 to 0.9527 when 224×224 images were replaced by 320×320 images for pretraining. Similar results can be found in Dataset2. The accuracy, AUC, sensitivity and specificity can be enhanced up to 92.60%, 0.9803, 0.973 and 0.9263, respectively. In this study, 0.9803 and 0.973 are the highest values achieved by ViT-Large in terms of AUC and sensitivity, respectively. However, the increment of the image size from 224×224 to 320×320 cannot significantly improve the performance of the model with randomly initialized weights. This is also true for models pre-trained on ImageNet. It may be ascribed to the larger error in interpolation.

**Table 5. The pre-training information in ViT-Base.**

| Dataset | Image size | Epoch | Masking Ratio |
|---------|-----------|-------|---------------|
| DataSet1 | 224×224 | 100, 200, 300, 400, 500, 600, 700, 800 | 0.75 |
| | 320×320 | 100, 200, 300, 400, 500, 600, 700, 800, 900, 1000 | 0.75 |
| | 448×448 | 100, 200, 300, 400, 500, 600, 700, 800, 900, 1000 | 0.75 |
| DataSet2 | 224×224 | 100, 200, 300, 400, 500, 600, 700, 800 | 0.75 |
| | 320×320 | 100, 200, 300, 400, 500, 600, 700, 800, 900, 1000 | 0.75 |
| | 448×448 | 100, 200, 300, 400, 500, 600, 700, 800, 900, 1000 | 0.75 |

**Table 6. The results of ViT-Large with masking ratio of 0.75.**

| Pre-training Dataset | Image size | Masking Ratio | Accuracy | AUC | Sensitivity | Specificity |
|---|---|---|---|---|---|---|
| - | 224×224 | - | 84.38% | 0.9038 | 0.8581 | 0.8341 |
| | 320×320 | - | 82.74% | 0.9126 | 0.8581 | 0.8065 |
| ImageNet1k | 224×224 | 0.75 | **93.15%** | 0.9801 | 0.8919 | **0.9585** |
| | 320×320 | 0.75 | 91.23% | 0.9745 | 0.8581 | 0.9493 |
| DataSet1 | 224×224 | 0.75 | 89.59% | 0.9624 | 0.9324 | 0.9124 |
| | 320×320 | 0.75 | 90.41% | 0.9719 | 0.9527 | 0.9078 |
| DataSet2 | 224×224 | 0.75 | 90.68% | 0.9738 | 0.9121 | 0.9171 |
| | 320×320 | 0.75 | 92.60% | **0.9803** | **0.9730** | 0.9263 |

Moreover, pre-training model with different types of images (natural and retinal images) leads to classification variations of the model. As shown in Table 6, ViT-Large pre-trained on natural images exhibits lower sensitivity compared with that on retinal images. And the sensitivity of ViT-Large pre-trained on natural images is much lower than the specificity. While, ViT-Large pre-trained on retinal images yields the opposite results. That is, the sensitivity of ViT-Large pre-trained on retinal images is higher than the specificity.

The influences of masking ratio on the performance of ViT-Large are shown in Table 7. MAE with 50% masking ratio slightly improves the performance of the model compared to that with 75% masking ratio. This conclusion is similar to that by Cai et al. [36]. However, it should be pointed out that the enhancement by increasing image size (224×224 to 320×320) is larger than that by reducing the masking ratio.

The results of the ViT-Base experiments are shown in Table 8. The performance of different models on images larger than 224×224 (320×320 and 448×448) were examined and compared with each other. It can be found that the performance of ViT-Base is similar to that of ViT-Large.

Compared with the model using 224×224 images of Dataset1 for pretraining, the accuracy with 320×320 and 448×448 images can be increased by 1.92% and 2.46%, respectively. And the model accuracy based on 17,000 images with the size of 448×448 is 92.05%, which exceeds all the results of ViT-Base using ImageNet pre-trained weights in this study. Using more than 100,000 images with the size of 320×320 in Dataset2 for pretraining, ViT-Base achieves the highest accuracy of 93.42% in this study. However, further expanding the image size to 448×448 does not significantly improve the accuracy, the AUC can be increased to 0.9853 and exceeded the results obtained by other models in this study. The sensitivity of ViT-Base pre-trained on natural and retinal images is higher than the specificity, which is different from that of ViT-Large.

It can be found that ViT-Base using the ImageNet pre-trained weights could improve the model performance when the input image size was increased from 224×224 to 320×320. This is because that large-size images provide more details and show a major impact on model performance. However, when the input image size was increased from 320×320. to 448×448, both

**Table 7. The results of ViT-Large with different masking ratio.**

| Pre-training Dataset | Image size | Masking Ratio | Accuracy | AUC | Sensitivity | Specificity |
|---|---|---|---|---|---|---|
| DataSet1 | 224×224 | 0.75 | 89.59% | 0.9624 | **0.9324** | 0.9124 |
| | 224×224 | 0.5 | 90.96% | 0.9705 | 0.9257 | **0.9217** |
| DataSet2 | 224×224 | 0.75 | 90.68% | 0.9738 | 0.9121 | 0.9171 |
| | 224×224 | 0.5 | **91.78%** | **0.9768** | **0.9324** | **0.9217** |

**Table 8. The results of ViT-Base with masking ratio of 0.75.**

| Pre-training Dataset | Image size | Masking Ratio | Accuracy | AUC | Sensitivity | Specificity |
|---|---|---|---|---|---|---|
| - | 224×224 | - | 84.38% | 0.9086 | 0.9122 | 0.7972 |
| | 320×320 | - | 85.21% | 0.9145 | 0.8784 | 0.8341 |
| | 448×448 | - | 83.29% | 0.9202 | 0.9122 | 0.7788 |
| ImageNet1k | 224×224 | 0.75 | 90.96% | 0.978 | 0.9257 | 0.8986 |
| | 320×320 | 0.75 | 92.05% | 0.9806 | 0.9257 | 0.9171 |
| | 448×448 | 0.75 | 90.41% | 0.9718 | 0.9460 | 0.8756 |
| DataSet1 | 224×224 | 0.75 | 89.59% | 0.9669 | 0.9054 | 0.8940 |
| | 320×320 | 0.75 | 91.51% | 0.9720 | 0.9324 | 0.9355 |
| | 448×448 | 0.75 | 92.05% | 0.9761 | 0.9595 | 0.9355 |
| DataSet2 | 224×224 | 0.75 | 92.05% | 0.9805 | 0.9460 | 0.9217 |
| | 320×320 | 0.75 | **93.42%** | 0.9825 | **0.9662** | **0.9539** |
| | 448×448 | 0.75 | 93.15% | **0.9853** | **0.9662** | 0.9447 |

model accuracy and AUC decrease. It may be attributed to that the weight interpolation show more significant effect on model performance than that of image size. It should be pointed out that the model with pretraining could lead to a significant improvement in model performance. However too many pretraining epochs lead to overfitting, which is consistent with the finds by El-Noub et al. [33] and Zhou et al. [35].

## Comparison to state-of-the-art methods

We compared the classification performance of VMLRI for rDR on the APTOS dataset with other state-of-the-art models, including CNN and Bayesian Neural Networks (BNN). The comparative results are presented in Table 9. Zhang et al. [43] proposed a multi-model domain adaptation (MMDA) method, and they trained it on source domains including DDR, IDRiD, Messidor, and Messidor-2 datasets and tested it on the target domain APTOS. Their method achieved high sensitivity but had an accuracy of 90.6%, which is lower than that of VMLRI (93.42%). Zhang et al. [44] introduced the Source-Free Transfer Learning (SFTL) method with a similar concept of domain adaptation. They considered the EyePACS as the source domain and trained the model on this dataset. After training, the model was tested on the APTOS. Compared to MMDA, SFTL exhibited higher accuracy but slightly lower sensitivity. In this study, VMLRI outperforms SFTL in terms of accuracy, sensitivity, and specificity.

Vives-Boix et al. [45] tested the classification performance of VGG16 and InceptionV3 on the APTOS dataset. In terms of sensitivity, VMLRI surpasses VGG16 and InceptionV3. While, in accuracy, VMLRI outperforms VGG16 but is slightly inferior to InceptionV3. Inspired by metaplasticity in the field of neuroscience, they proposed AmInceptionV3 by improving

**Table 9. Performance comparison with state-of-the-art diabetic retinopathy classification methods on the APTOS dataset.**

| Method | Accuracy | AUC | Sensitivity | Specificity |
|---|---|---|---|---|
| MMDA [43] | 90.6% | - | **0.985** | - |
| SFTL [44] | 91.2% | - | 0.951 | 0.858 |
| VGG16 [45] | 92.91% | - | 0.94 | - |
| InceptionV3 [45] | 93.59% | - | 0.93 | - |
| AmInceptionV3 [45] | **94.46%** | - | 0.9 | - |
| BNN [46] | - | 0.961 | - | - |
| VMLRI | 93.42% | **0.9825** | 0.9662 | **0.9539** |

InceptionV3, which achieved an accuracy of 94.46%. However, the sensitivity of AmInceptionV3 was reduced to be 0.9. Apart from CNN, Jaskari et al. [46] attempted to classify rDR using BNN, obtaining an AUC of 0.961 on the APTOS dataset. This result is lower than VMLRI's AUC (0.9825), and it is noteworthy that they used retinal images with a higher resolution of 512×512. In conclusion, the proposed VMLRI shows high sensitivity and specificity and a competitive AUC of 0.9825, and demonstrates balanced performance compared with state-of-the-art models on.

## Discussion

If DR can be detected and treated at an early stage, further damage to vision caused by DR can be effectively prevented. Thus, a novel VMLRI has been proposed for the classification of rDR. It has been found that the classification performance of the present model pre-trained with MAE on more than 100,000 large-size (320×320 and 448×448) fundus retinal images at 75% masking ratio is better than that with the weights from ImageNet. Although the pretraining with a masking ratio of 50% provides a slight improvement in model performance, it consumes more computing resources. Furthermore, ViT-Large and ViT-Base indicate similar results in accuracy and AUC, and they show advantages in sensitivity and specificity, respectively. Therefore, the ViT-Base pre-trained with a masking ratio of 75% is recommended, and the masking ratio and network architecture can be adjusted based on the results and situation.

This study shows that MAE can provide more flexibility to the input image and substantially reduce the number of images required for pretraining of downstream tasks. However, the required computing resources increases rapidly as the size of input image increases. For the self-attention mechanism in the Transformer, the computational complexity is quadratic with respect to the input sequence length, and thus the training time during the pre-training phase will be significantly extended if the image resolution increases. However, this issue can be mitigated by setting a higher masking rate to reduce the sequence length. It should be also noted that the proposed method exhibits a degree of overfitting, which was also reported by El-Noub et al. [33] and Zhou et al. [35].

The present VMLRI model is beneficial to incorporate the domain knowledge of downstream tasks and modify the network architecture, instead of limiting the network structure in order to use pre-training weights. In order to further enhance the pre-trained model's classification performance, the Transformer network structure will be modified to incorporate relevant prior knowledge from DR domain in the future. In addition, introducing DR-related regression or detection tasks on top of the MAE image reconstruction task can be also taken into account.

## Acknowledgments

We thank the reviewers for their insightful comments that helped improve our manuscript's overall quality.

## Author Contributions

**Conceptualization:** Yaoming Yang.

**Formal analysis:** Peng Xu.

**Funding acquisition:** Peng Xu.

**Investigation:** Peng Xu.

**Methodology:** Yaoming Yang.

**Project administration:** Shuxia Qiu.

**Resources:** Peng Xu.

**Software:** Zhili Cai.

**Supervision:** Shuxia Qiu.

**Validation:** Yaoming Yang, Zhili Cai, Shuxia Qiu.

**Writing – original draft:** Yaoming Yang.

**Writing – review & editing:** Peng Xu.

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
