## [Decision Letter · Decision Letter 0]

2 Nov 2023

PONE-D-23-29847Vision transformer with masked autoencoders for referable diabetic retinopathy classification based on large-size retina imagePLOS ONE

Dear Dr. Xu,

Thank you for submitting your manuscript to PLOS ONE. After careful consideration, we feel that it has merit but does not fully meet PLOS ONE’s publication criteria as it currently stands. Therefore, we invite you to submit a revised version of the manuscript that addresses the points raised during the review process.

We look forward to receiving your revised manuscript.

Kind regards,

Yawen Lu, Ph.D

Academic Editor

PLOS ONE

4. We note that Figure 1 in your submission contain copyrighted images. All PLOS content is published under the Creative Commons Attribution License (CC BY 4.0), which means that the manuscript, images, and Supporting Information files will be freely available online, and any third party is permitted to access, download, copy, distribute, and use these materials in any way, even commercially, with proper attribution. For more information, see our copyright guidelines: http://journals.plos.org/plosone/s/licenses-and-copyright.

Additional Editor Comments:

Comments to the Author:

In conclusion, after careful review and consideration of the paper titled "Vision transformer with masked autoencoders for referable diabetic retinopathy classification based on large-size retina image", the initial decision is major revision. This decision is based on the consensus of the reviewers and the following concerns and contributions of the paper:

From R1, the authors should compare the method with other state-of-the-art CNN models, state the novelty and contribution of the proposed work, the limitations of the method, and other details in implementation and result analysis.

From R2, the authors should make the architecture clear. R2 also mentions some issues in the writing and layout in tables, contributions and figures.

Reviewers' comments:

Reviewer's Responses to Questions

**Comments to the Author**

1. Is the manuscript technically sound, and do the data support the conclusions?

Reviewer #1: Partly

Reviewer #2: Yes

2. Has the statistical analysis been performed appropriately and rigorously? 

Reviewer #1: No

Reviewer #2: Yes

3. Have the authors made all data underlying the findings in their manuscript fully available?

Reviewer #1: No

Reviewer #2: Yes

4. Is the manuscript presented in an intelligible fashion and written in standard English?

Reviewer #1: Yes

Reviewer #2: No

5. Review Comments to the Author

Reviewer #1: The author implemented the article entitled "Vision transformer with masked autoencoders for referable diabetic retinopathy classification based on large-size retina image". There are many queries need to be addressed.

1. Did the authors compared the Vision transformer performance with any other state of art CNN?

2. Since already many researchers implemented Vision Transformer in Diabetic retinopathy database. Then what is the novelty of the proposed work?

3. Literature review given in the related work is not sufficient . Need to add some more recent papers.

4. Training and testing data spilt?

5. Whether 5 fold or 10 fold validation carried out?

6. Information about the hyperparameters used.

7. Have you performed error analysis?

8. Methods need to be elaborated.

9. Discussion is incomplete. The author should discuss the results of their proposed work with existing related literature works.

10. Include the limitations and future scope of the work.

11. Result analysis need to be improved in Result section.

Reviewer #2: 1. while talking about datasets : start with the APTOS dataset as this was the one mentioned rigorously within the introduction, then move forward to the other datasets.

2. In the introduction: explain what DeiT-S and other techniques stand for as you added the details for CNN and ViT.

3. Confusion between the closing paragraphs within the introduction and the related work. why dont you move all this part to be combined with the related work. Divide the related work to work conducted on standard size retinal images and larger retina images in order to present the drawbacks mentioned within the introduction based on solid facts .

4. Figures needs more labeling between arrows in order to better understand the sequence.

5. Move the contributions to the conclusions, replace those with objectives instead to enhance the readability.

6. For the results tables, highlight the best performing techniques through all datasets and all evaluation criterea.

7.The architecture is very unclear, every component needs to be explained separately with the motivation behind it and what it adds to the overall technique.

6. PLOS authors have the option to publish the peer review history of their article (what does this mean?). If published, this will include your full peer review and any attached files.

Reviewer #1: No

Reviewer #2: No

---

## [Author Response · Author response to Decision Letter 0]

27 Nov 2023

We have carefully revised our manuscript according to the journal requirements and comments by editor and reviewers.We have highlighted the revised text in blue type in our revised manuscript. And, we have provided response to each comment, uploaded separated “Response to Reviews”.

---

## [Decision Letter · Decision Letter 1]

29 Jan 2024

PONE-D-23-29847R1Vision transformer with masked autoencoders for referable diabetic retinopathy classification based on large-size retina imagePLOS ONE

Dear Dr. Xu,

Thank you for submitting your manuscript to PLOS ONE. After careful consideration, we feel that it has merit but does not fully meet PLOS ONE’s publication criteria as it currently stands. Therefore, we invite you to submit a revised version of the manuscript that addresses the points raised during the review process.

We look forward to receiving your revised manuscript.

Kind regards,

Yawen Lu

Academic Editor

PLOS ONE

Journal Requirements:

Additional Editor Comments:

Please address the minor comments on the diagram, logic flow and literature reviews. After the modification, the manuscript should be ready to publish.

Reviewers' comments:

Reviewer's Responses to Questions

**Comments to the Author**

1. If the authors have adequately addressed your comments raised in a previous round of review and you feel that this manuscript is now acceptable for publication, you may indicate that here to bypass the “Comments to the Author” section, enter your conflict of interest statement in the “Confidential to Editor” section, and submit your "Accept" recommendation.

Reviewer #1: All comments have been addressed

Reviewer #2: (No Response)

2. Is the manuscript technically sound, and do the data support the conclusions?

Reviewer #1: Yes

Reviewer #2: Yes

3. Has the statistical analysis been performed appropriately and rigorously? 

Reviewer #1: Yes

Reviewer #2: Yes

4. Have the authors made all data underlying the findings in their manuscript fully available?

Reviewer #1: Yes

Reviewer #2: Yes

5. Is the manuscript presented in an intelligible fashion and written in standard English?

Reviewer #1: Yes

Reviewer #2: No

6. Review Comments to the Author

Reviewer #1: The authors have addressed the reviewer comments. Hence the article can be accepted in its current form.

Reviewer #2: Most of the comments have been covered (greatest thanks for your effort), however I will reiterate two comments that will make the paper up to standard:

1- Diagrams to be clearer (especially ViT diagram), The logic flow is very unclear without reading the text

2- Dividing the literature review into sub sections based on the size of the dataset worked on

7. PLOS authors have the option to publish the peer review history of their article (what does this mean?). If published, this will include your full peer review and any attached files.

Reviewer #1: No

Reviewer #2: No

---

## [Author Response · Author response to Decision Letter 1]

1 Feb 2024

We appreciate the comments from the reviewers. We have revised Figures 1 and 2 by adding annotations to make them clear. Additionally, we have revised the related work section, and added subsections to improve readability.

---

## [Decision Letter · Decision Letter 2]

8 Feb 2024

Vision transformer with masked autoencoders for referable diabetic retinopathy classification based on large-size retina image

PONE-D-23-29847R2

Dear Dr. Xu,

We’re pleased to inform you that your manuscript has been judged scientifically suitable for publication and will be formally accepted for publication once it meets all outstanding technical requirements.

Kind regards,

Yawen Lu, Ph.D

Academic Editor

PLOS ONE

Additional Editor Comments (optional):

Dear authors:

Regarding your submission:

PONE-D-23-29847R2

Vision transformer with masked autoencoders for referable diabetic retinopathy classification based on large-size retina image

We have received feedbacks from the previous reviewers and are announcing that your work has been Accepted for publication in PLOS ONE.

Please follow the following steps and provide a camera-ready version of your manuscript. Congratulation again!

Reviewers' comments:

Reviewer's Responses to Questions

**Comments to the Author**

1. If the authors have adequately addressed your comments raised in a previous round of review and you feel that this manuscript is now acceptable for publication, you may indicate that here to bypass the “Comments to the Author” section, enter your conflict of interest statement in the “Confidential to Editor” section, and submit your "Accept" recommendation.

Reviewer #2: All comments have been addressed

2. Is the manuscript technically sound, and do the data support the conclusions?

Reviewer #2: Yes

3. Has the statistical analysis been performed appropriately and rigorously? 

Reviewer #2: Yes

4. Have the authors made all data underlying the findings in their manuscript fully available?

Reviewer #2: Yes

5. Is the manuscript presented in an intelligible fashion and written in standard English?

Reviewer #2: Yes

6. Review Comments to the Author

Reviewer #2: Comments have been addressed, no further comments are required to be detailed.

Language was fixed, organizational structure and diagrams were revised

7. PLOS authors have the option to publish the peer review history of their article (what does this mean?). If published, this will include your full peer review and any attached files.

Reviewer #2: No

---

## [Editor Report · Acceptance letter]

26 Feb 2024

PONE-D-23-29847R2 

PLOS ONE

Dear Dr. Xu, 

I'm pleased to inform you that your manuscript has been deemed suitable for publication in PLOS ONE. Congratulations! Your manuscript is now being handed over to our production team.

Kind regards, 

on behalf of

Dr. Yawen Lu 

Academic Editor

PLOS ONE